# Gallic Acid: A Natural Phenolic Compound Exerting Antitumoral Activities in Colorectal Cancer via Interaction with G-Quadruplexes

**DOI:** 10.3390/cancers14112648

**Published:** 2022-05-26

**Authors:** Victoria Sanchez-Martin, María del Carmen Plaza-Calonge, Ana Soriano-Lerma, Matilde Ortiz-Gonzalez, Angel Linde-Rodriguez, Virginia Perez-Carrasco, Inmaculada Ramirez-Macias, Marta Cuadros, Jose Gutierrez-Fernandez, Javier Murciano-Calles, Juan Carlos Rodríguez-Manzaneque, Miguel Soriano, Jose Antonio Garcia-Salcedo

**Affiliations:** 1GENYO, Centre for Genomics and Oncological Research, Pfizer/University of Granada/Andalusian Regional Government, 18016 Granada, Spain; victoria.sanchez@genyo.es (V.S.-M.); mcarmen.plaza@genyo.es (M.d.C.P.-C.); ana.soriano@genyo.es (A.S.-L.); matilde.ortiz@genyo.es (M.O.-G.); angel.linde@genyo.es (A.L.-R.); virginia.perez@genyo.es (V.P.-C.); inmaculada.ramirez@genyo.es (I.R.-M.); mcuadros@ugr.es (M.C.); juancarlos.rodriguez@genyo.es (J.C.R.-M.); msoriano@ual.es (M.S.); 2Microbiology Unit, Biosanitary Research Institute IBS-Granada, University Hospital Virgen de las Nieves, 18014 Granada, Spain; josegf@ugr.es; 3Department of Biochemistry, Molecular Biology III and Immunology, University of Granada, 18016 Granada, Spain; 4Department of Physiology, University of Granada, 18011 Granada, Spain; 5Centre for Intensive Mediterranean Agrosystems and Agri-Food Biotechnology (CIAIMBITAL), University of Almeria, 04001 Almeria, Spain; 6Department of Physical Chemistry, Unit of Excellence for Chemistry Applied to Biomedicine and the Environment, and Institute of Biotechnology, University of Granada, 18071 Granada, Spain; jmurciano@ugr.es

**Keywords:** G-quadruplex, phenol, gallic acid, cancer, colorectal cancer

## Abstract

**Simple Summary:**

Gallic acid, a natural phenolic compound in diet, interacts with DNA G-quadruplexes both in vitro and in vivo. In particular, gallic acid targets G-quadruplexes in ribosomal DNA and *CMYC* oncogene, affecting gene expression. This action leads to antitumoral effects in colorectal cancer. In a patient cohort with CRC, we demonstrate that gallic acid could be explored as a therapeutic agent.

**Abstract:**

Natural phenolic compounds have gained momentum for the prevention and treatment of cancer, but their antitumoral mechanism of action is not yet well understood. In the present study, we screened the antitumoral potential of several phenolic compounds in a cellular model of colorectal cancer (CRC). We selected gallic acid (GA) as a candidate in terms of potency and selectivity and extensively evaluated its biological activity. We report on the role of GA as a ligand of DNA G-quadruplexes (G4s), explaining several of its antitumoral effects, including the transcriptional inhibition of ribosomal and *CMYC* genes. In addition, GA shared with other established G4 ligands some effects such as cell cycle arrest, nucleolar stress, and induction of DNA damage. We further confirmed the antitumoral and G4-stabilizing properties of GA using a xenograft model of CRC. Finally, we succinctly demonstrate that GA could be explored as a therapeutic agent in a patient cohort with CRC. Our work reveals that GA, a natural bioactive compound present in the diet, affects gene expression by interaction with G4s both in vitro and in vivo and paves the way towards G4s targeting with phenolic compounds.

## 1. Introduction

Colorectal cancer (CRC) is the third most common cancer accounting for approximately 10% of all annually diagnosed cancers and cancer-related deaths worldwide [1]. In the development of CRC, multiple mutations or epigenetic changes are accumulated, leading to the transformation of non-tumoral colonic mucosa into colonic adenocarcinoma, subsequent carcinoma, and metastasis [1]. Although the molecular drivers of CRC have been described to a great extent, treatment options have been slightly developed. CRC treatment includes surgery, radiotherapy, and neoadjuvant and palliative chemotherapies [1]. However, these strategies have had a limited impact on cure rates and long-term survival, most of them causing negative side effects.

Both hereditary and environmental risk factors play a part in the development of CRC. Incidence rates of CRC fluctuate worldwide, with the highest incidences found in developed countries, whichcould be explained by differences in eating and cultural habits [2]. Among the main, largely modifiable environmental factors are exercise, smoking, excessive alcohol intake, and diet [2]. In this regard, whilst consumption of red and processed meats increases the risk of developing CRC, diets enriched in fruits, vegetables, and fibers are proposed to reduce the risk of disease onset and progression [3]. Natural bioactive compounds present in food, especially phenolic compounds, possess important biological properties such as anticancer, anti-inflammatory, and antioxidant activities [4]. Phenolic compounds are secondary metabolites produced in plants and microorganisms with a common aromatic ring bearing at least one hydroxyl group. More than 8000 natural phenolic compounds have been identified to date [5]. The antitumoral efficacy of phenolic compounds differs according to variations in their structure as well as their molecular target [6]. In this work, we are focused on five different phenolic compounds, including resveratrol (RSV), piceid (PIC), tyrosol (TYR), hydroxytyrosol (HTYR), and gallic acid (GA). All of them are easily available in a regular diet. A plethora of studies support the idea that anticancer properties of phenolic compounds comprise scavenging free radicals, induction of enzymes involved in xenobiotics metabolism, modulation of gene expression, and regulation of cellular signaling pathways, including those involved in cell proliferation and invasion [7]. Interestingly, phenolic compounds are accumulated in the cell nucleus rather than in any other organelles [8]. However, the exact molecular mechanism underlying many of their actions in CRC models is yet to be fully clarified. 

G-quadruplexes (G4s) are considered promising therapeutic targets in cancer. G4s are four-stranded, noncanonical secondary structures formed via Hoogsteen hydrogen-bonding of four guanines in planar quartets thatself-stack [9]. Interestingly, G4s participate in key genome functions such as transcription, replication, genome stability, and epigenetic regulation, displaying numerous connections to cancer biology [10]. In the present study, we aimed to screen the antitumoral activity of several phenolic compounds in a CRC progression cellular model. Next, we extensively evaluated the biological activity of GA as an antitumoral candidate both in vitro and in vivo. According to our results, GA may constitute an approach for CRC treatment in the future.

## 2. Materials and Methods

### 2.1. Phenolic Compounds

Five phenolic compounds easily available in a regular diet, such as resveratrol, piceid, tyrosol, hydroxytyrosol, and gallic acid, were screened according to theirantitumoral activity. All of them were acquired from Merck (Darmstadt, Germany) (R5010 for resveratrol; 15,721 for piceid; PHL80166 for tyrosol; H4291 for hydroxytyrosol; G7384 for gallic acid). Stock solutions were prepared in dimethyl sulfoxide (DMSO) at 10 mM and stored at −20 °C. For some experiments, 10 µM CX5461 (HY-13323, MedChemExpress, Monmouth Junction, NJ, USA), a well-known G-quadruplex ligand, was used as a positive control. Stock solutions of CX5461 were prepared in DMSO at 1 mM and stored at −20 °C.

### 2.2. Cell Culture

CRL1790, SW480 and SW620 cell lines were purchased from American Type Culture Collection (ATCC). Cells were cultured at 37 °C with 5% CO_2_ atmosphere in medium supplemented with 10% fetal bovine serum (10270106, Gibco, Waltham, MA, USA), 10 mg/mL penicillin, 10 mg/mL streptomycin, 100 mg/mL amphotericin, and 0.03% L-Glutamine as recommended by the ATCC. MEM medium (M5650, Merck, Darmstadt, Germany) was used for non-tumoral CRL1790, and RPMI 1640 medium (L0501-500, Biowest, Nuaillé, France) was used for both tumoral SW480 and SW620 cell lines.

### 2.3. Patient Samples 

This study was approved by the local Ethical Committee of the University of Granada (Granada, Spain). Samples of patients with CRC were obtained from University Hospital Virgen de las Nieves (Granada, Spain), and informed consent was obtained from all of them. Tumor biopsies from 15 patients and histologically non-tumoral adjacent tissue from 7 patients were collected before treatment and freshly frozen until RNA extraction. The patient cohort was homogeneous. In addition, gene expression data from Oncomine database were subjected to bioinformatic analyses. In particular, “TCGA Colorectal” dataset with non-tumoral (N = 22) and colon adenocarcinoma (N = 101) samples was used with the following filters: (1) “Cancer Type: Colorectal Cancer”; (2) “Gene: *POLR1A*/*CMYC*”; (3) “Data Type: mRNA”; (4) “Analysis Type: Cancer vs. Normal Analysis”, and (5) “Threshold Setting Condition (*p* < 0.001, fold change > 2, gene rank = top 10%)”.

### 2.4. Cytotoxic Assay

Cytotoxic activity was screened using Resazurin Fluorimetric Assay (R7017, Merck, Darmstadt, Germany), according to the manufacturer’s protocol. Cells seeded into 96-well plates (8 × 10^3^ cells/well) were treated for 48 h with phenolic compounds at increasing concentrations from 2 × 10^−5^ µM to 100 µM. Negative control with vehicle DMSO was included. Fluorescence was determined using Nanoquant Infinite M200 Pro multi-plate reader (Tecan). Half-maximal inhibitory concentration (IC_50_) values were determined in triplicate by non-linear regression with Graphpad (Prism).

### 2.5. Cell Cycle Analysis

Cell cycle analyses were carried out by flow cytometry with propidium iodide (PI) (P4864, Merck, Darmstadt, Germany). Cells (10^6^) were seeded into 10 cm culture dishes and treated with GA IC_50_ for 24 h or with the vehicle DMSO as control. Cells were then fixed with ice-cold 70% ethanol on ice and stained with 0.04 mg/mL PI and 0.1 mg/mL ribonuclease A (19101, Qiagen, Hilden, Germany). Cell cycle distribution was determined by an analytical DNA flow cytometer (FACSVerse, BD Biosciences) with instrument settings on low mode and FlowJo software v10. 

### 2.6. Immunofluorescence Assays 

Cells were seeded on 13 mm circular coverslips and placed in 24-well plates. After exposure to different experimental conditions, fixation was performed with 4% (*v*/*v*) paraformaldehyde (P6148, Merck, Darmstadt, Germany) for 10 min at room temperature (RT), permeabilization with 0.1% (*v*/*v*) Triton-X100 (T8787, Merck, Darmstadt, Germany) for 10 min and blocking with 10% bovine serum albumin (A7906, Merck, Darmstadt, Germany) containing 0.5% (*v*/*v*) Triton-X100 for 30 min at RT. Primary antibodies were incubated for 1 h at RT and secondary antibodies for 30 min at 4 °C in darkness. Finally, all coverslips were mounted onto slides (J2800AMNZ, Thermo scientific, Waltham, MA, USA), including DAPI (4′,6-diamidino-2-phenylindole) for nuclear counterstain. Images were acquired on a Confocal Zeiss LSM 710 inverted microscope with a 63× immersion objective. In a different manner, BG4 immunofluorescence was conducted as previously described [11]. BG4 mean nuclear fluorescence intensity was quantified using Fiji software (N > 250). Antibodies used are listed in Appendix A.

### 2.7. qRT-PCR

Total cellular RNA was isolated from different experimental group cells using Trizol Reagent (15596, Invitrogen, Waltham, MA, USA). Reverse transcription was conducted using RevertAid First Strand cDNA Synthesis Kit (K1622, Thermo Scientific, MA, USA) according to manufacturer’s protocol with random primers. Quantitative PCR was performed with SYBR Green (4309155, Thermo Scientific, MA, USA) on 7900HT Fast Real-time PCR System (Applied Biosystems). Target mRNA levels were normalized to actin (ΔCt), and fold change was determined using the 2^−ΔΔCt^ method. Experiments were conducted in triplicate. Primers used for this study are listed in Appendix A.

### 2.8. Western Blot Analyses

Protein extract from different experimental conditions was extracted using RIPA lysis buffer containing 1% PMSF (P7626, Merck, Darmstadt, Germany), 1% protease inhibitor cocktail (PIC) (P8340, Merck, Darmstadt, Germany), and 1% sodium orthovanadate (S6508, Merck, Darmstadt, Germany). Quantification of protein levels was achieved by Bradford method following manufacturer’s protocol (500-0006, BioRad, Hercules, CA, USA). Protein content was loaded on 12% SDS-polyacrylamide gel (1610148, BioRad, Hercules, CA, USA) for electrophoresis and wet transferred to nitrocellulose membranes (66485, Pall corporation, New York, NY, USA). Membranes were blocked with 5% semi-skimmed milk and incubated overnight at 4 °C with antibodies for γH2AX and actin as housekeeping. Then, membranes were incubated with horseradish peroxidase-labeled antibodies for 1 h at RT. After luminal solution (1705060, BioRad, Hercules, CA, USA) incubation, chemiluminiscence signals were measured using Image Quant LAS 4000 (GE Healthcare Life Sciences). Experiments were performed in triplicate, and representative images are shown. Protein levels were quantified by ImageJ. Antibodies used are listed in Appendix A.

### 2.9. G4-Oligonucleotides Prefolding

G4-oligonucleotides listed in Appendix A were purchased from Integrated DNA Technologies (Clareville, IA, USA). All of them were dissolved in G4s buffer (10 mM potassium phosphate, 100 mM potassium chloride at pH 7.0), heated at 95 °C for 10 min, slowly cooled to RT, and stored at 4 °C.

### 2.10. Fluorescent Intercalator Displacement (FID) Assay

We used TOPRO3 (T3605, Thermo Scientific, MA, USA) as a fluorescent intercalator for FID assays. In particular, 5 mM TOPRO3 was incubated with 10 mM prefolded G4s and exposed to 10 µM GA in 96-well plates. TOPRO3 was excited at 642 nm, and emission profile was monitored between 650–800 nm with Infinite M200 Plate Reader (Tecan). Fluorescence values were calculated as follows: %Fluorescence = A/B × 100; where (A) is the fluorescence value in presence of GA, and (B) corresponds to the fluorescence value in GA-free controls. All assays were conducted in triplicate. The G4-oligonucleotides used in the current study are listed in Appendix A.

### 2.11. PCR-Stop Assay

Sequences of the test G4-oligonucleotides and the corresponding partially complementary oligonucleotides used in PCR-stop assays are listed in Appendix A. The reactions were performed in 1× PCR Combination buffer, containing 20 pmol of each pair of oligonucleotides, 0.2 mM dNTPs, 2.5 U Hot Start *Taq* polymerase (733–1331, VWR, Radnor, PA, USA), and increasing amounts of GA from 0 µM to 100 µM. PCR products were amplified in a Veriti Thermal Cycler (Applied Biosystems) with the following cycling conditions: 95 °C for 15 min, followed by 30 cycles of 95 °C for 30 s, 58 °C for 30 s, and 72 °C for 30 s. Amplified products were resolved on 3% agarose gel in 1× TBE (100 mM Tris base, 100 mM boric acid, 2 mM EDTA) and stained with GelGreen (41005, Biotium, Fremont, CA, USA). Gel Image was analyzed on ImageQuant LAS 4000. Three independent reactions were conducted per concentration, and representative lanes were displayed.

### 2.12. Circular Dichroism (CD) Spectra

CD spectra were recorded at 25 °C on a JASCO 715 CD spectropolarimeter in G4s buffer conditions (10 mM potassium phosphate buffer containing 100 mM potassium chloride at pH 7.0). The concentration of the prefolded G4 DNA was 10 μM, and GA was added at 100 μM and incubated overnight to register the new spectrum. The used wavelength range was 230–700 nm with 100 nm/min as scan speed. The cuvette path length was 0.1 cm, and three accumulation spectra were averaged for each measurement. The G4-oligonucleotides used in the current study are listed in Appendix A.

### 2.13. Ultraviolet-Visible (UV-Vis) Titration

UV-vis absorption spectra were registered in a Varian Cary 50 UV-vis spectrophotometer at 25 °C. Concentration of the prefolded G4 DNA was 5 μM in G4s buffer (10 mM potassium phosphate buffer containing 100 mM potassium chloride at pH 7.0). Once the DNA was placed in the cuvette, a concentrated solution of GA (1 mM) was routinely added, 1 μL each time, with a Hamilton syringe and subsequently mixed with a pipette. After each addition of the GA solution, a UV-vis spectrum was recorded. In total, 10 μL of GA solution were added, with a final ratio of 1:20 G4 DNA:GA. For the blank, the same experiment with the successive additions was repeated, beginning just with buffer in the cuvette. Then, each titration spectrum was subtracted from its corresponding blank. The path length of the cuvette was 0.3 cm and the wavelength range used was 235–320 nm. Experiments were conducted in triplicate. The G4-oligonucleotides used in the current study are listed in Appendix A. Dissociation constants (K_d_) were evaluated at 265 nm using the following equation:(1)$$A=A_{m}+\left(A_{ml}-A_{m}\right)\times \frac{\left(M+L+K_{d}\right)-\sqrt{{\left(M+L+K_{d}\right)}^{2}-4\times M\times L}}{2\times M}$$
where A is the absorbance signal, A_m_ is the signal of DNA in absence of the ligand, A_ml_ is the signal of DNA in presence of the ligand, M is the total concentration of DNA, L is the total concentration of ligand, and K_d_ is the dissociation constant.

### 2.14. Xenograft Studies

NOD scid gamma (NSG) mice were purchased from CIBM-UGR and housed at the animal facility according to institutional guidelines (Approved Ethical Committee #152-CEEA-OH-2016). For xenograft generation, 1 × 10^5^ SW480 cells in 100 μL PBS were subcutaneously injected into the flank of 8-week-old female mice. Treatment started when tumors reached ~20 mm^3^. Mice were randomly divided into two groups of seven mice each and treated with either vehicle DMSO or 200 mg/kg of GA intraperitoneally every other day for 38 days. Animals were monitored every two days after cell injection until final time point when they were sacrificed, and tumors were dissected for further analyses. Specifically, tumor volumes were determined every two days using digital calipers according to the formula: In progress tumor volume = (π × length × width^2^)/6 [12]. 

### 2.15. Immunohistochemical Analysis of Tumor Sections 

Fixation, paraffin-embedding, and sectioning of tumor samples were performed by the histopathology core service at the Centre for Genomics and Oncological Research (Granada, Spain). In order to evaluate the percentage of proliferating cells, tumor sections were immunostained with Ki67 and counterstained with hematoxylin and eosin (H&E) at Atrys Health (Barcelona, Spain). The staining was visualized using NDP.view2 Viewing software (Hamamatsu), and Ki67 coverage was quantified on ten different images per tumor using Fiji software. In order to measure BG4 signal, tumor sections were dewaxed and rehydrated following standard methods. Epitope retrieval was performed at 100 °C for 20 min with citrate buffer (citrate-based buffer pH 6.0) according to previous studies [13]. After blocking, staining was achieved with BG4 antibody overnight at 4 °C, following a 1 h incubation with anti-FLAG at RT and a 30 min incubation with an anti-mouse antibody at RT in darkness. Slides were then counterstained for 5 min with DAPI to visualize the cell nuclei. Antifade Mowiol (81381, Merck, Darmstadt, Germany) was used as mounting media. Images were acquired on a Confocal Zeiss LSM 710 inverted microscope with a 63× immersion objective. BG4 mean nuclear fluorescence intensity was quantified using Fiji software (N > 2000). Antibodies used are listed in Appendix A.

### 2.16. Statistical Analysis

Statistical significance was assessed using Student’s two-tailed *t*-test. Values represent mean ± standard deviation. For all tests, *p*-values below 0.05 were considered significant and expressed as follows: * *p* < 0.05; ** *p* < 0.01 and *** *p* < 0.001.

## 3. Results

### 3.1. Gallic Acid Shows Anticancer Activity In Vitro

Natural products such as phenolic compounds have recently attracted significant attention for their anticancer properties. In order to identify potential drugs for CRC, we determined the cytotoxic activity of five natural phenols, including resveratrol (RSV), piceid (PIC), tyrosol (TYR), hydroxytyrosol (HTYR), and gallic acid (GA). We used three different cell lines to mimic CRC progression. In particular, CRL1790 are colon epithelial cells simulating the non-tumoral stage. In addition, SW480 is Dukes’ type B colorectal adenocarcinoma cells representing the primary tumor. Finally, SW620 constitutes Dukes’ type C colorectal adenocarcinoma derived from metastasic site cells to mimic the most aggressive metastasic stage. Interestingly, SW480 and SW620 were established from the same patient. Cytotoxic potency upon 48 h treatment with phenolic compounds was examined by determination of half-maximal inhibitory concentration (IC_50_) values using the Resazurin method (Table 1). Only GA inhibited cell growth in SW480 and SW620 at clinically-relevant concentrations (IC_50_ values lower than 30 µM) [14]. Furthermore, GA exhibited a higher selectivity for cancer cells with a minimal affectation of non-tumoral CRL1790 cells (IC_50_ values higher than 100 µM) (Figure 1A). Based on these results, GA is a lead compound for CRC treatment and was selected for further studies.

### 3.2. Gallic Acid Induces Cell Cycle Arrest and Nucleolus Disintegration

GA (3,4,5-trihydroxybenzoic acid) is a naturally occurring triphenolic compound with low molecular weight (Figure 1B). GA is widely present in the plant kingdom and largely found in different food sources [15]. According to its cytotoxic activity (shown above), cell cycle analysis was performed on SW480 cells by FACS with propidium iodide staining (Figure 1C). After GA IC_50_ treatment for 24 h, the frequency of cells at different stages of the cell cycle changed with respect to non-treated cells (45.34% of cells at G1, 34.16% at S, 7.65% at G2/M). GA induced a prominent S and G2/M phases increase (28.30% of cells at G1, 43.82% at S, 14.46% at G2/M). These results suggest that GA might alter DNA replication triggering S and G2/M phase cell cycle arrest, but GA binding to DNA has not been previously reported. To explore this hypothesis, we determined the intracellular localization of nucleolar proteins by immunofluorescence of SW480 cells treated with GA IC_50_ for 6 h because we had previously observed that some DNA-binding compounds affect nucleolar organization [16]. In particular, we analyzed three proteins that are indicative of nucleolus status, such as nucleolin (NCL), fibrillarin (FBL), and Pol I catalytic subunit A (POLR1A) (Figure 1D). GA altered their nucleolar localization, including translocation of NCL from the nucleolus to the nucleoplasm and segregation of FBL to nucleolar periphery caps. However, significant changes in POLR1A were not observed. Altogether these results imply that GA causes a remarkable cell cycle arrest and nucleolar stress. 

### 3.3. Gallic Acid Stabilizes G-Quadruplexes Inhibiting the Transcription of Several Oncogenes and Induces DNA Damage

Nucleolus disintegration is a hallmark of ribosomal RNA transcription blockage by some compounds targeting DNA. Specifically, these compounds bind to G4s and are shown to modulate transcription [17]. In particular, transcription of several oncogenes (including *BCL2*, *CMYB*, *CMYC*, *KRAS*, *VEGFA*) is thought to be controlled by stabilization of G4s [18], and ribosomal DNA gene also harbors G4 sequences which impair ribosomal RNA synthesis [16]. In this context, we aimed to analyze whether GA affects the transcription of G4-enriched genes by qRT-PCR. In the case of ribosomal DNA, we measured the short-lived 5′ external transcribed spacer (*5′ETS*) of the pre-RNA, whichreflects the ribosomal RNA synthesis rate [19]. Treatment of SW480 cells with GA IC_50_ for 6 h resulted in a significant downregulation of several genes which contain G4s (Figure 2A). The well-known G4 ligand, CX5461, also influenced the expression levels of some of these genes. Next, we investigated the G4-stabilization properties of GA in SW480 cells by immunofluorescence with the G4 selective antibody BG4 [11]. GA IC_50_ induced a notorious and significant increase in nuclear BG4 signal after treatment for 6 h, suggesting that GA strongly trapped G4 structures (Figure 2B,C). A similar effect was observed using CX5461 as a positive control. Then, we investigated DNA damage response because the induction of double-strand breaks is a well-known G4s stabilization associated effect [20]. To this end, we measured the phosphorylation of histone H2AX on Ser-139 (γH2AX), a DNA damage marker, by Western Blot. In the same way that occurred upon exposure to CX5461, GA IC_50_ incubation for 6 h significantly induced DNA damage (Figure 2D,E and Appendix A). Therefore, GA acts as a G4 ligand inducing the downregulation of several G4-enriched oncogenes and DNA damage.

### 3.4. Gallic Acid Interacts with G4s in 5′ETS and CMYC 

The transcriptional inhibition of genes containing G4s in their promoters by GA prompted us to examine whether GA interacted with some of these G4s using the TOPRO3 fluorescent intercalator displacement (FID) assay. This assay is based on the displacement of a DNA light-up probe (TOPRO3) from G4 DNA upon competition with G4-binding ligands [21]. For that, we used G4 sequences which were fully characterized in previous studies such as *5′ETS* [16], *BCL2* [22], *CMYB* [23], *CMYC* [23], *KRAS* [23], *VEGFA* [24] and telomeric sequences [25]. GA 10 µM exhibited significant binding to G4s found in *5′ETS* and in the promoter of *CMYC*, decreasing the fluorescence percentage down to 84.2% and 84.3%, respectively (Figure 3A). The stabilization of these G4 structures by GA was further investigated by a PCR-stop assay with the test oligonucleotide, including the target G4 sequence and a partially complementary oligonucleotide. The specific binding of ligands with intramolecular G4 structures blocks the progression of the DNA polymerase, and the final double-stranded PCR product is not detected. In this regard, GA inhibited the accumulation of amplified products when added to PCR reactions, including G4 sequences for *5′ETS* and *CMYC* at 100 µM and 50 µM, respectively (Figure 3B and Appendix A). In contrast, after DMSO (vehicle) treatment at the same dilution asGA, no inhibition was observed even at the highest concentration. In order to understand the effect of GA on the *5′ETS* and *CMYC* G4 conformation, circular dichroism (CD) studies were performed. Both G4s showed a positive band around 260 nm and a negative band at 240 nm, indicating the existence of a parallel G4 conformation. Upon addition of 100 µM GA, the intensity of the positive CD band of both *5′ETS* and *CMYC* G4s decreased (Figure 3C) in a similar way as is described elsewhere [26]. However, the variations of the CD profile were minimal, thus proving that the overall folding of these G4s was preserved even upon ligand binding. Moreover, the CD pattern is used for the determination of binding modes with DNA [27]. The absence of induced CD signal (Appendix A) for any of the tested G4s suggests a mode of binding weaker than intercalation (i.e., end-stacking or electrostatic interaction), as reported for other G4 ligands [28,29]. Further binding studies were carried out using UV-vis spectroscopy. When GA was added to the *5′ETS* G4, the complex peaked around 254 nm and displayed hypochromicity, and when added to *CMYC* G4, the UV-vis absorption spectra exhibited both hypochromicity in the peak at 254 nm together with an isosbestic point at 297 nm, indicating the existence of two different species in equilibrium with each other (Figure 3D). Data analysis rendered a dissociation constant (K_d_) of 148 µM and 113 µM for *5′ETS* and *CMYC* G4s, respectively, corroborating that GA is a weak binder for both G4s. Altogether, these results confirm that GA interacts with *5′ETS* and *CMYC* G4s.

### 3.5. Gallic Acid Blocks Tumor Progression and Stabilizes G4 Structures In Vivo

We investigated the activity of GA in a mouse xenograft model of CRC to determine if the in vitro activity of GA was paralleled in vivo. According to previous reports with SW480 cells [30], we executed xenograft studies by subcutaneous injection in NOD scid gamma (NSG) mice. Intraperitoneal treatment started when tumors reached ~20 mm^3^ (35 days post-injection), and a therapeutic schedule with either a vehicle or 200 mg/kg of GA, every other day for 38 days, was explored based on previous studies [31]. Initially, each group included seven animals but two mice from the control group were excluded because the tumors had developed late. Therefore, five mice were considered in the control group and seven in the GA-treated one. No body weight reduction or adverse effects such as tumor ulceration was observed at any time during the study. Over the course of the experiment, GA caused a robust blockade on the progression of tumor xenografts in treated mice compared to the vehicle control group (Figure 4A). In particular, when comparing the tumor volume between control and treated animals at each time point by Student’s two-tailed *t*-test, significant differences were observed from the 23rd day after initiation of the treatment. Histopathological analyses of tumors from animals sacrificed at the ending point were also conducted. Immunofluorescence analysis with BG4 showed a significant increase in nuclear BG4 signal in tumors from GA-treated animals (Figure 4B,C), which confirmed that GA also had a direct action at G4 sites in vivo. Furthermore, immunohistochemical assessment of the antigen Ki67 was used to estimate cell proliferation. The results demonstrated that the coverage of Ki67 was significantly decreased in tumors originated in GA-treated mice, indicating that tumors are less proliferative after treatment with GA (Figure 4D,E). Finally, gene expression data obtained by qRT-PCR showed that, upon GA treatment, *5′ETS* and *CMYC* were significantly downregulated in responder mice (Figure 4F,G). Altogether our data strongly suggest that GA stabilizes G4 structures in vivo leading to an inhibition of tumor growth in CRC xenografts. 

### 3.6. Gallic Acid Could Be Explored for Patients with CRC as Therapy 

Cancer cells overexpress ribosomal machinery [32] and *CMYC* [33] to meet their requirements for limitless proliferation. Therefore, downregulation of *5′ETS* and *CMYC* by GA could be a feasible strategy for CRC treatment. In order to assess the translational potential of GA for patients with CRC, we determined *POLR1A* (that transcribes ribosomal gene) and *CMYC* expression levels by qRT-PCR in a patient cohort with CRC. In particular, 15 CRC tumoral samples together with seven colorectal biopsies derived from non-tumoral adjacent tissue were examined. The analysis was performed by the ΔCt method, which shows relative gene expression using actin as housekeeping gene. As expected, ΔCt *POLR1A* and ΔCt *CMYC* decreased in CRC, meaning that both *POLR1A* and *CMYC* were overexpressed in colorectal tumors compared with non-tumoral tissues (Figure 5A). These expression patterns were further validated in a larger cohort of patients with CRC through bioinformatic analyses from the Oncomine database. The expression levels of *POLR1A* and *CMYC,* measured by log2 median-centered ratios, were significantly higher in colon adenocarcinoma than that in the non-tumoral tissues (Figure 5B). These findings reveal that *POLR1A* and *CMYC* constitute clinically attainable targets in CRC, positioning GA as a candidate for CRC treatment that requires translational exploration in the future. 

## 4. Discussion

It is suggested in studies that a diet rich in fruits and vegetables could reduce the incidence of CRC. This effect has been mostly attributed to phenolic compounds [34]. Apart from chemoprevention, the development of anticancer therapies involving natural phenols has undergone exponential growth in recent years [35]. However, the underlying mechanisms of phenolic compounds are not fully understood [36]. Here, we have studied the antitumorigenic potential in the CRC of five different phenolic compounds. Among them, we have selected GA as a promising candidate and provide detailed evidence of its mechanism of action via binding to G4s.

GA is a phenolic molecule widely present in varied food sources, with a mean content of 1.75 mg/100 g and 8.25 mg/100 g in different fruits and vegetables, respectively [15]. Among other phenols that we have tested, we selected GA due to its potent and selective antitumoral effect in CRC. Such difference in efficacy seems to be due to the variations in their chemical structure. In fact, compounds with a greater number of hydroxylic groups exhibited better anticancer activity compared to those with a lower number. In this regard, GA, which possesses three hydroxyl groups attached to three, four, and five positions of a benzoic acid core, has been reported to be more effective than other phenols [7]. Moreover, it has been shown that GA suppresses cell growth not only in CRC [37] but also in other types of cancer [38]. 

In agreement with previous studies demonstrating that phenolic compounds are associated with the dysregulation of the cell cycle [39], our results indicated that GA induces an arrest at S and G2/M phases. However, our work includes relevant findings in this regard. Firstly, we identified that GA causes nucleolar stress, and secondly, that GA induces downregulation of G4-containing genes. These effects prompted us to further examine the role of GA as a G4-ligand. Thirdly, we confirmed that GA stabilizes G4s in a cellular environment both in vitro and in xenograft sections by immunofluorescence with BG4 antibody. In accordance withits G4-stabilizing properties, GA shares some effects with other well-established G4 ligands. Consistent with a previous study [40], our results indicated that GA induces DNA damage. In fact, many G4-stabilizing ligands produce DNA damage in the vicinity of G4-forming sequences [20]. Therefore, to the best of our knowledge, our work is the first demonstration that a natural phenol binds to G4s in human cancer cell lines, paving the way for future studies.

In addition, we demonstrated GA binding to G4s present in *5′ETS* and in the promoter of *CMYC* through biophysical studies. Based on the most simplistic model, G4s are considered repressors of transcription by preventing polymerase processivity [41]. Hence, GA, once inside the cell, would bind to the G4 found in *5′ETS* and *CMYC*, which could explain the downregulation of *5′ETS* and *CMYC* upon GA treatment that we observed both in vitro and in vivo. Strikingly, G4s harbored in *5′ETS* and *CMYC* have in common that both adopt a parallel structure, and both are biologically relevant substrates of nucleolin, the most abundant nucleolar phosphoprotein [23]. However, although GA also inhibited the transcription of other G4-enriched oncogenes such as *BCL2*, *CMYB*, *KRAS,* and *VEGFA,* we were not able to identify what G4s were involved in these regulation loops, and further investigation is required. In this sense, compelling research has suggested that G4s may not only be involved in proximal transcriptional control but also part of long-distance epigenetic mechanisms [42]. Therefore, we must consider the G4 not as an isolated entity within a specific genomic location but instead as a part of an interconnected network of interactions with other biomolecules in living cells [42].

Interestingly, the anticancer activities of GA have been extensively disclosed in the literature before. In CRC, most studies have mainly attributed its anticancer effects to the generation of reactive oxygen species and induction of apoptosis [37,43,44]. In addition, GA inhibited angiogenesis through suppression of VEGF secretion in ovarian cancer [45]. Moreover, GA imposed anti-inflammatory effects on prostate cancer through inhibition of the expression of many cytokines such as IL-6 [46]. However, altogether these experiments were carried out after exposure to GA for a long time (24, 48, or even 72 h). Based on our observations, we suggest that part of the previously reported effects may be considered as downstream and indirect processes that derive from the behavior of GA as a G4-stabilizing ligand. Furthermore, epigenome-modifying abilities of GA have been observed in tobacco-associated cancers where GA reduced DNA methyltransferases activity within one week [47]. It is possible that such effects are explained since GA increases the percentage of stabilized G4s, and these structures themselves mold the DNA methylome by sequestering DNA methyltransferase 1 (DNMT1) [48]. Still, how those processes are so carefully orchestrated within the cells through G4 targeting with GA requires further investigation. 

From a translational point of view, we demonstrate that patients with CRC overexpress *POLR1A* and *CMYC*, and thus, we propose that G4-mediated downregulation of ribosomal and *CMYC* genes exerted by GA would constitute an attainable approach for CRC treatment. On behalf of our in vivo experiments, treatment with GA successfully reduces tumor growth in CRC xenografts without causing observable damage to major organs. Thereupon, simultaneous targeting of multiple pathways by G4 stabilization results in an advantageous approach for CRC treatment, although the variability of potency and selectivity among different G4s and pathways remain unclear. Notwithstanding, polyphenols often display a poor bioavailability when administered as pure active principles, constituting an important limit to their use [35]. Their bio-transformation, at the colon level, by the heterogeneity of human gut microbiota, also leaves open enormous spaces for further research [49]. In particular, two bacterial strains specifically produce GA in humans [50]. However, the bioavailability of these compounds can be improved by their administration in nanotechnology-based formulations or even in combination with other phytochemicals [51]. Moreover, the possibility of combining conventional chemotherapeutic drugs with polyphenols provides valuable advantages, such as the increase in efficiency and the reduction of side effects [34].

Undoubtedly, our work stands out the implication of nutrigenomics in cancer treatment. Nutrigenomics is focused on the existing reciprocal interactions between genes and nutrients at a molecular level [52]. Here, we reveal how a natural bioactive compound that we consume in our regular diet, GA, is able to affect gene expression by interaction with G4s. The stabilization properties of GA are inferred from experiments with cells in vitro and, most importantly, with animals, being possibly extrapolated to humans. Through this underlying mechanism, GA is directly involved in nutrigenomics, which ultimately governs human health and disease.

## 5. Conclusions

We demonstrate that the natural phenolic compound GA is a G4-binding small molecule, and we provide detailed evidence of its mechanism of action with in vitro cell assays and in vivo models for human CRC.

## Figures and Tables

**Figure 1 cancers-14-02648-f001:**
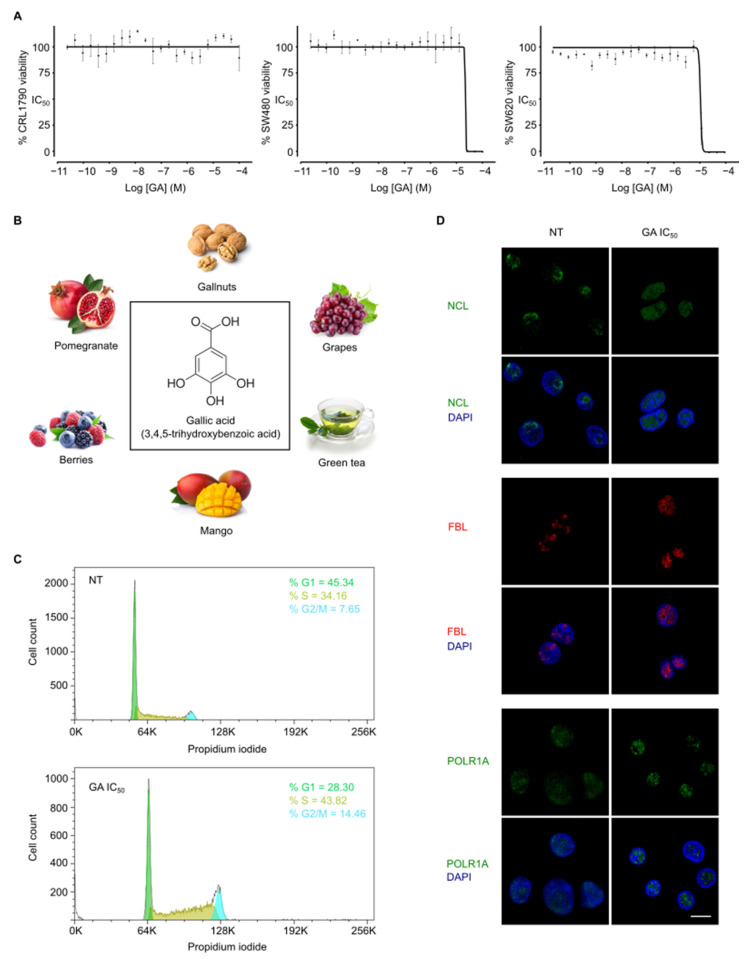
Gallic acid induces cell cycle arrest and nucleolus disintegration. (**A**) Dose response curve of GA for non-tumoral CRL1790, tumoral SW480 and metastatic SW620 cells after treatment during 48 h. (**B**) Chemical structure of GA selected as antitumoral candidate and natural sources where it can be found. (**C**) Histograms of SW480 cells treated with DMSO (non-treated, NT) or treated with GA IC_50_ for 24 h in propidium iodide flow cytometry analysis. (**D**) Immunofluorescence images of SW480 cells treated with vehicle DMSO (NT) or treated with GA IC_50_ for 6 h, and stained for nucleolin (NCL), fibrillarin (FBL) or POLR1A. Merged images with DAPI for DNA counterstaining are also shown. Scale bar, 10 µm.

**Figure 2 cancers-14-02648-f002:**
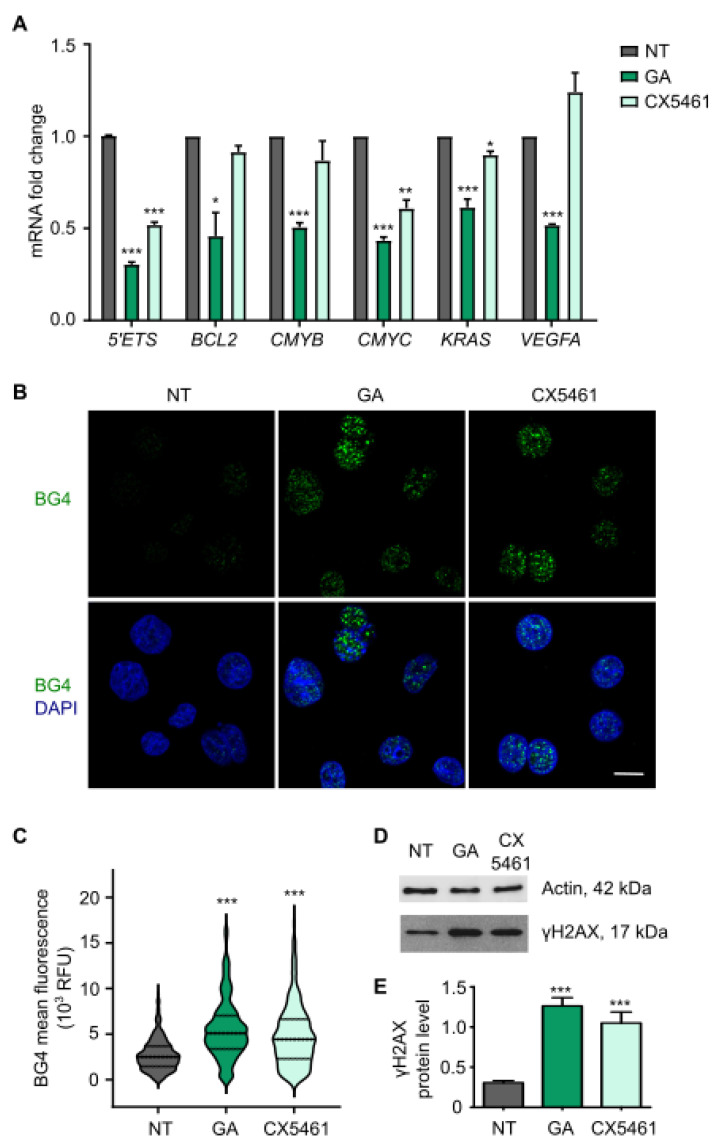
Gallic acid stabilizes G4s inhibiting the transcription of several oncogenes and inducing DNA damage. (**A**) SW480 cells were treated with vehicle DMSO (NT), GA IC_50_ or CX5461 10 µM for 6 h and transcription of different G4-enriched genes was analyzed by qRT-PCR. Columns represent mean ± standard deviation. (**B**) Immunofluorescence images of SW480 cells treated with vehicle (NT), GA IC_50_ or CX5461 10 µM for 6 h and stained with the G4-selective antibody, BG4. Merged images with DAPI for DNA counterstaining are also shown. Scale bar, 10 µm. (**C**) Quantification of nuclear BG4 mean fluorescence intensity by Fiji analysis from cells in (**B**) (N > 250). (**D**) Western blot experiments in SW480 cells upon treatment with vehicle (NT), GA IC_50_ or CX5461 10 µM for 6 h to determine protein levels of γH2AX as a marker of DNA damage and actin as housekeeping gene. (**E**) Quantification of γH2AX protein levels normalized to actin of data in (**D**) by ImageJ. Columns represent mean ± standard deviation. For all tests, *p*-values below 0.05 were considered significant and expressed as follows: * *p* < 0.05; ** *p* < 0.01 and *** *p* < 0.001. Please see uncropped Western blot in Appendix A.

**Figure 3 cancers-14-02648-f003:**
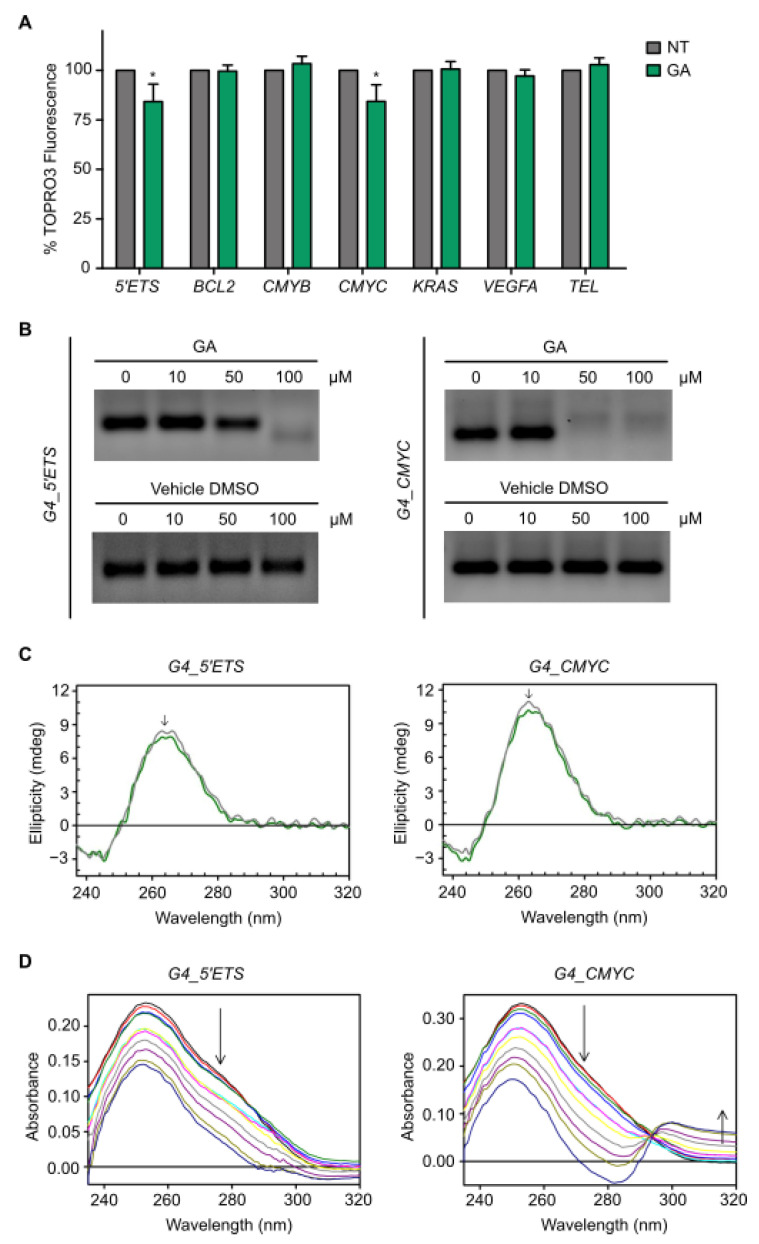
Gallic acid interacts with G4s in *5′ETS* and *CMYC.* (**A**) FID assay using different G4-containing oligonucleotides to determine the TOPRO3 fluorescence percentage in the absence (NT) or presence of GA 10 µM. Columns represent mean ± standard deviation. (**B**) Effect of increasing concentrations of GA or the corresponding vehicle DMSO on a PCR-stop assay including the G4-containing oligonucleotide of *5′ETS* and *CMYC*. (**C**) CD spectra obtained with the G4s formed by *5′ETS* and *CMYC* in the absence or presence of GA 100 µM. The arrows indicate the direction of movement of CD peaks upon addition of GA. (**D**) UV-vis spectra of the G4s formed by *5′ETS* and *CMYC* upon the addition of increments of GA 10 µM up to 100 µM as final concentration. The arrows indicate the direction in which the absorption peak moves after interaction of GA with G4. For all tests, *p*-values below 0.05 were considered significant and expressed as follows: * *p* < 0.05. Please see uncropped gels in Appendix A.

**Figure 4 cancers-14-02648-f004:**
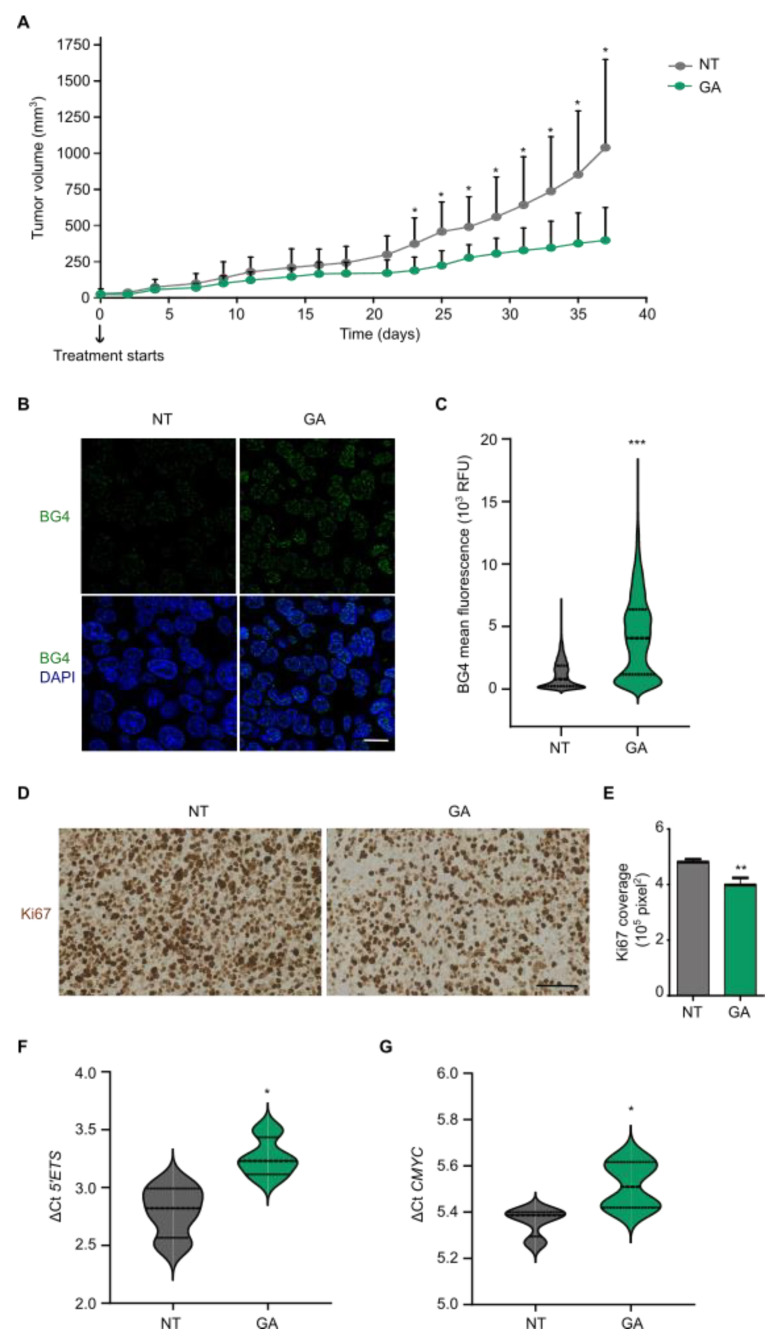
Gallic acid blocks tumor progression and binds to G4s in vivo. (**A**) Tumor volume of SW480 xenograft mice treated with DMSO vehicle control (N = 5) or 200 mg/kg GA (N = 7) every other day for 38 days. Each point represents the mean tumor volume of the group ± standard deviation (only the positive bars are shown). (**B**) Representative images of BG4 immunofluorescence in tumor sections from control and GA-treated xenografts. Merged images with DAPI for DNA counterstaining are also shown. Scale bar, 10 µm. (**C**) Quantification of nuclear BG4 mean fluorescence intensity by Fiji analysis from tumor sections in (**B**) (N > 2000). (**D**) Representative images of Ki67 staining in tumor sections from control and GA-treated mice. Hematoxylin and eosin were used as counterstaining. Scale bar, 100 µm. (**E**) Quantification of Ki67 coverage from tumor sections in (**D**) by Fiji analysis (ten different images per tumor). (**F**) ΔCt results for *5′ETS* expression obtained by qRT-PCR from control and responder mice. (**G**) ΔCt results for *CMYC* expression obtained by qRT-PCR from control and responder mice. For all tests, *p*-values below 0.05 were considered significant and expressed as follows: * *p* < 0.05; ** *p* < 0.01 and *** *p* < 0.001.

**Figure 5 cancers-14-02648-f005:**
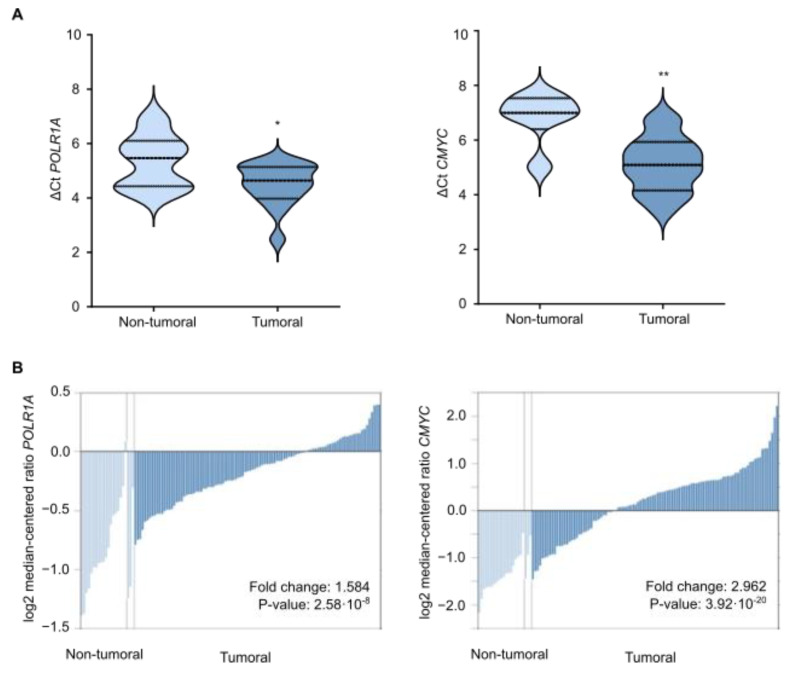
Gallic acid could be explored for patients with CRC as therapy. (**A**) ΔCt results for *POLR1A* and *CMYC* obtained by qRT-PCR in the cohort of patients with CRC. (**B**) Analysis of *POLR1A* and *CMYC* expression levels in Oncomine database with “TCGA Colorectal” dataset including non-tumoral (N = 22) and tumoral (N = 101) samples. For all tests, *p*-values below 0.05 were considered significant and expressed as follows: * *p* < 0.05 and ** *p* < 0.01.

**Table 1 cancers-14-02648-t001:** IC_50_ values for phenolic compounds in the cellular model of CRC. IC_50_ values represent phenols concentration inhibiting cell growth by 50% and are expressed as mean ± standard deviation. Experiments were performed in biological triplicates.

Phenolic Compound	Cell Line	IC_50_ (µM)
Resveratrol	CRL1790	>100
	SW480	>100
	SW620	>100
Piceid	CRL1790	>100
	SW480	>100
	SW620	>100
Tyrosol	CRL1790	>100
	SW480	>100
	SW620	>100
Hydroxytyrosol	CRL1790	>100
	SW480	>100
	SW620	71.94 ± 3.52
Gallic acid	CRL1790	>100
	SW480	22.39 ± 2.12
	SW620	11.83 ± 1.54

## Data Availability

Not applicable.

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
