# Peer review of "Gallic Acid: A Natural Phenolic Compound Exerting Antitumoral Activities in Colorectal Cancer via Interaction with G-Quadruplexes"

_cancers, 2022, doi:10.3390/cancers14112648_

Round 1

Reviewer 1 Report

The manuscript, entitled “Gallic acid: a natural phenolic compound exerting antitumoral activities in colorectal cancer via interaction with G-quadruplexes” by Sanchez-Martin and al. describes as the gallic acid affects gene expression by interaction with G-quadruplex both in vivo and in vitro.  This is reflected on a antitumoral effects in colorectal cancer.  The study demonstrated that gallic acid causes cell cycle arrest, nucleolar stress and induction of DNA damage in the same way of other G-quadruplex ligands. The authors claimed the stabilization of G-quadruplexes by gallic acid with xenograft model of CRC.

Comment: This study was performed using a combination of experimental techniques from cytotoxic assay to fluorescence, CD, UV and others. The present study is significant and the experimental work appears to be acceptable and completed, but and some points have to be clarified:

-The authors should perform melting experiment by CD in order to compare the melting temperature of the free quadruplexes (5’ETS and CMYC) with the melting temperature of the complexes G-quadruplex/gallic acid in order to confirm that the ligand stabilizes the quadruplex. The CD results reported prove the preservation of the G-quadruplex structure upon interaction with the ligand. 

Did the induced CD (ICD) signals appear in the visible region after treatment with gallic acid? As ICD signals are usually related to intercalation, the absence of ICD suggests a mode of binding weaker than intercalation, such as end-stacking or electrostatic interaction. About this they should introduce in references: a) White, E.W.; Tanious, F.; Ismail, M.A.; Reszka, A.P.; Neidle, S.; Boykin, D.W.; Wilson, W.D. “Structure-specific recognition of quadruplexDNAby organic cations: Influence of shape, substituents and charge” . Biophys. Chem. 2007, 126, 140–153; b) L. Musso, S. Mazzini, A.  Rossini et al. ”c-MYC G-quadruplex binding by the RNA polymerase I inhibitor BMH-21 and analogues revealed by a combined NMR and biochemical Approach” Biochimica et Biophysica Acta,  20181862, 615–629  https://doi.org/10.1016/j.bbagen.2017.12.002; c) C. Platella, S. Mazzini, E. Napolitano et al. “Plant-Derived Stilbenoids as DNA-Binding Agents: From Monomers to Dimers”. Chem. Eur. J. 2021, 27, 1–15. doi.org/10.1002/chem.202101229. Chemistry - A European Journal (2021), 27(34), 8832-8845.

-Can they evaluate the binding constants?

In conclusion the manuscript may be favourably considered for publication in Cancers if the authors will consider my comments.

Author Response

Thanks for your comments. Undoubtedly, we consider that our manuscript has been improved with these changes. Please, see attached our point-by-point response. 

Reviewer 2 Report

In this interesting manuscript, the author explored the anti-tumor activity of a phenolic compound Gallic acid. They have shown Gallic acid exerts anti-cancer effect by binding to DNA G-quadruplexes; therefore, cell cycle arrest, nucleolar stress, and DNA damage. Consequently, tumor cell cytotoxicity occurs both in vivo and in vitro conditions. The article is well written, organized, and adequately describes methods and discussion. I have some minor comments:

  1. Include the dose repose curve for the Gallic acid in Fig. 1.
  2. Line 271-273 may transfer to the introduction section.
  3. 1c is a bit hazy; if possible, change with higher resolution and clearer pictures.
  4. How did author perform statistical analysis for change in tumor volume? Try to do with other statistical analyses e.g. ANOVA
  5. There are some minor spelling or grammatical mistakes: line 62 anti-inflammatory, line 386 tumor ulceration was, line 388 treated mice, line 409 hematoxylin and eosin were, line 443 phenols have, line 495 epigenome-modifying

Author Response

(The authors gave the same response as above.)

Round 2

Reviewer 1 Report

OK